# WIGNER KERNELS: BODY-ORDERED EQUIVARIANT MACHINE LEARNING WITHOUT A BASIS

## ABSTRACT

Machine-learning models based on a point-cloud representation of a physical object are ubiquitous in scientific applications and particularly well-suited to the atomic-scale description of molecules and materials. Among the many different approaches that have been pursued, the description of local atomic environments in terms of their discretized neighbor densities has been used widely and very successfully. We propose a novel density-based method which involves computing "Wigner kernels". These are fully equivariant and body-ordered kernels that can be computed iteratively at a cost that is independent of the basis used to discretize the density and grows only linearly with the maximum body-order considered. Wigner kernels represent the infinite-width limit of feature-space models, whose dimensionality and computational cost instead scale exponentially with increasing order of correlations. We present several examples of the accuracy of models based on Wigner kernels in chemical applications, for both scalar and tensorial targets, reaching state-of-the-art accuracy on the popular QM9 benchmark dataset. We discuss the broader relevance of these findings to equivariant geometric machine-learning.

## 1 INTRODUCTION

Machine-learning techniques are widely used to perform tasks on 3D objects, from pattern recognition and classification to property prediction (Gumhold et al., 2001; Guo et al., 2021; Wu et al., 2019; Li et al., 2021). In particular, different flavors of geometric machine learning (Bronstein et al., 2021) have been used widely in applications to chemistry, biochemistry and condensed-matter physics (Gainza et al., 2020; Carleo et al., 2019; Ceriotti et al., 2021). Given the coordinates and types of atoms seen as a decorated point cloud, ML models act as a surrogate for accurate electronic-structure simulations, predicting all types of atomic-scale properties that can be obtained from quantum mechanical calculations (Behler & Parrinello, 2007; Bartók et al., 2010; Rupp et al., 2012; Gilmer et al., 2017; Brockherde et al., 2017; Ceriotti, 2022). These include scalars such as the potential energy, but also vectors and tensors, which require models that are covariant to rigid rotations of the system (Bereau et al., 2015; Glielmo et al., 2017).

In this context, body-ordered models have emerged as an elegant and accurate way of describing how the behavior of a molecule or a crystal arises from a hierarchy of interactions between pairs of atoms, triplets, and so on – a perspective that has also been widely adopted in the construction of traditional physics-based interatomic potentials (Finnis & Sinclair, 1984; Horsfield et al., 1996; Medders et al., 2015; Sanchez et al., 1984). By only modeling interactions up to a certain body order, these methods generally achieve low computational costs. Futhermore, since low-body-order interactions are usually dominant, focusing machine-learning models on their description also leads to excellent accuracy and data-efficiency. Several body-ordered models have been proposed for atomistic machine learning; while most work has focused on simple linear models (Drautz, 2019; Dusson et al., 2022; Nigam et al., 2020), it has been shown that several classes of equivariant neural networks (Thomas et al., 2018a; Anderson et al., 2019; Batzner et al., 2022a) can be interpreted in terms of the systematic construction of hidden features that are capable of describing body-ordered symmetric functions (Nigam et al., 2022a; Batatia et al., 2022a). Kernel methods are also popular in the field of atomistic chemical modeling (Bartók et al., 2010; Rupp et al., 2012; Chmiela et al., 2017a; Faber et al., 2018; Grisafi et al., 2018; Glielmo et al., 2018), as they provide a good balance between the simplicity of linear methods and the flexibility of non-linear models. In most cases,

they are used in an invariant setting, and the kernels are manipulated to incorporate higher-order terms in a non-systematic way (Deringer et al., 2021). Even though the body-ordered construction is particularly natural for chemical applications, the formalism can be applied equally well to any point cloud (Thomas et al., 2018b), and so it is also relevant for more general applications of geometric learning.

In this work, we present an approach to build body-ordered equivariant kernels in an iterative fashion. Crucially, the iterations only involve kernels themselves, entirely avoiding the definition of a basis to expand the radial and chemical descriptors of each atomic environment and the associated scaling issues. The resulting kernels can be seen as the infinite-basis or infinite-width limit of many of the aforementioned equivariant linear models and neural networks. We demonstrate the excellent accuracy that is exhibited by these "Wigner kernels" in the prediction of scalar and tensorial properties, including the cohesive energy of transition metal clusters and high-energy molecular configurations, and both energetics and molecular dipole moments of organic molecules.

## 2 RELATED WORK

The definition of local equivariant representations of a point cloud is a problem of general relevance for computer vision (Marcon et al., 2022), but it is particularly important for atomistic applications, where the overwhelming majority of frameworks relies on the representation of atom-centered environments. Such atom-centered representations are often computed starting from the definition of local atomic densities around the atom of interest, which makes the predictions invariant with respect to permutation of atoms of the same chemical element. The locality of the atomic densities is often enforced via a finite cutoff radius within which they are defined, and it results in models whose cost scales linearly with the size of the system. The use of discretized atomic densities has also been linked to much increased computational efficiency in the evaluation of high-order descriptors, as they allow to compute them while avoiding sums over clusters of increasing order. This is sometimes referred to as the *density trick* (van der Oord et al., 2020; Musil et al., 2021).

**Smooth overlap of atomic positions and symmetry-adapted GPR**. The oldest model to employ the density trick is kernel-based SOAP-GPR (Bartók et al., 2013), which evaluates a class of 3-body invariant descriptors and builds kernels as their scalar products. Higher-body-order invariant interactions are generally included, although not in a systematic way, by taking integer powers of the linear kernels. This model has been used in a wide variety of applications (see Deringer et al. (2021) for a review). SA-GPR is an equivariant generalization of SOAP-GPR which aims to build equivariant kernels from "$\lambda$-SOAP" features (Grisafi et al., 2018). However, these kernels are built as products of a linear low-body-order equivariant part and a non-linear invariant kernel that incorporates higher-order correlations. As a result, these models are not strictly body-ordered, and they offer no guarantees of behaving as universal approximators (Pozdnyakov et al., 2020b).

**N-body kernel potentials**. In constrast, Glielmo et al. (2018) introduces density-based body-ordered kernels, and it proposes their analytical evaluation for low body orders. Nonetheless, these kernels are exclusively rotationally invariant, and the paper proposes a strategy based on approximate symmetrization as the only viable strategy to compute kernels of arbitrarily high body orders.

**MTP, ACE and NICE**. The moment tensor potential (MTP) (Shapeev, 2016) and the more recent, closely related atomic cluster expansion (ACE) (Drautz, 2019) and N-body iterative contraction of equivariants (NICE) (Nigam et al., 2022b) schemes consist of linear models based on a systematic hierarchy of equivariant body-ordered descriptors. These are obtained as discretized and symmetrized atomic density correlations, which are themselves simply tensor products of the atomic densities. Although several contraction and truncation schemes have been proposed (Willatt et al., 2018; Dusson et al., 2022; Darby et al., 2022), the full feature space of these constructions grows exponentially with the maximum body-order of the expansion.

**Equivariant neural networks**. Finally, equivariant neural networks (Thomas et al., 2018a; Anderson et al., 2019; Batzner et al., 2022a; Batatia et al., 2022b) have become ubiquitous in recent years, and they represent the state of the art on many atomic-scale datasets. Most, but not all (Musaelian et al., 2023), incorporate message-passing schemes. Equivariant architectures can be seen as a way to efficiently contract the exponentially large feature space of high-body-order density correlations (Nigam et al., 2022a). Even though the target-specific optimization of the contraction weights

gives these models great flexibility, they still rely on an initial featurization based on the expansion of the neighbor density on a basis and can only span a heavily contracted portion of the high-order correlations.

## 3 METHODS

**(Symmetry-adapted) Kernel ridge regression**. Throughout this work, we will employ Kernel ridge regression (KRR) to fit atomistic properties. In this context, kernel functions are defined between any two atomic-scale structures, so that the kernel $k(A, A')$ represents a similarity measure between structures $A$ and $A'$, where each structure is described via the set of positions and chemical elements of its atoms. As mentioned in Sec. 2, it is common practice – rooted in physical approximations (Prodan & Kohn, 2005) and usually beneficial to the transferability of the model (Musil et al., 2021) – to use atom-centered decompositions of the physical properties of a structure. This physical ansatz implies a kernel-mean-embedding (Muandet et al., 2017) form for the structure-wise kernels, that are decomposed into atom-pair contributions (De et al., 2016):

$$k(A, A') = \sum_{i \in A} \sum_{i' \in A'} k(A_i, A'_{i'}), \tag{1}$$

where $i$ runs over all atoms in structure $A$, $i'$ runs over all atoms in structure $A'$, and $A_i$, $A'_{i'}$ denote the atomic environments around atoms $i$ and $i'$, respectively. These are spherical neighborhoods of the central atom under consideration with radius $r_{\text{cut}}$ so that $A_i \equiv \{(a_j, \mathbf{r}_{ji})\}_{r_{ji} < r_{\text{cut}}}$ is a shorthand for all the Cartesian positions $\mathbf{r}_{ji}$ relative to the center, and the chemical element labels $a_j$, of the atoms within the cutoff radius $r_{\text{cut}}$.

As shown in Glielmo et al. (2017) and Grisafi et al. (2018), KRR can be extended to the prediction of atomistic properties that are equivariant with respect to symmetry operations in $SO(3)$ (3D-rotations $\hat{R}$). In order to build a symmetry-adapted model that is suitable for the regression of a property $y_\lambda^\mu$ (that transforms like the set $\{Y_\lambda^\mu\}_{\mu=-\lambda\ldots\lambda}$ of spherical harmonics of degree $\lambda > 0$ and order $-\lambda < \mu < \lambda$) it is sufficient to employ tensorial kernels $k_{\mu\mu'}^\lambda$, and a symmetry-adapted ansatz

$$\tilde{y}_\lambda^\mu(B) = \sum_A \sum_{\mu'} k_{\mu\mu'}^\lambda(B, A) \, c_A^{\mu'}, \tag{2}$$

where $c_A^{\mu'}$ are regression coefficients, $A$ is a structure in the training set, $B$ is a structure whose rotationally equivariant property $\tilde{y}_\lambda^\mu(B)$ is to be predicted, and the $k_{\mu\mu'}^\lambda$ kernels must obey

$$k_{\mu\mu'}^\lambda(\hat{R}A_i, \hat{R}'A'_{i'}) = \sum_{mm'} D_{\mu m}^\lambda(\hat{R}) D_{\mu' m'}^\lambda(\hat{R}') \, k_{mm'}^\lambda(A_i, A'_{i'}). \tag{3}$$

Here, $\mathbf{D}^\lambda(\hat{R})$ is the Wigner D-matrix associated with the rotation $\hat{R}$, i.e., the matrix representation of the rotation operator $\hat{R}$ in the basis of the irreducible representations of the $SO(3)$ group. In practice, most established invariant models use low-rank approximations of the kernel matrix, which result in a more favorable scaling with system size in training and predictions. See Deringer et al. (2021) for a recent review on kernel methods applied to atomistic problems.

**Atomic densities and body-ordered kernels**. As discussed in Sec. 2, a broad class of atomistic ML frameworks can be formulated in terms of discretized correlations of an atomic neighbor density defined within each environment (Willatt et al., 2019; Drautz, 2019; Musil et al., 2021; Nigam et al., 2022a). These are defined as scalar fields in real space $\rho_{i,a}(\mathbf{x})$, where $\mathbf{x} \in \mathbb{R}^3$, and given by

$$\rho_{i,a}(\mathbf{x}) = \sum_{j \in A_i, \, a_j = a} g(\mathbf{x} - \mathbf{r}_{ji}) \, f_{\text{cut}}(r_{ji}) \approx \sum_{nlm} c_{nlm}^a R_{nl}(x) Y_l^m(\hat{\mathbf{x}}). \tag{4}$$

Here, $j$ runs over all neighbors in $A_i$, $g$ is a three-dimensional Gaussian function, and $f_{\text{cut}}$ is a cutoff function which satisfies $f_{\text{cut}}(r \geq r_{\text{cut}}) = 0$, so that the $A_i$ neighborhoods are effectively restricted by a cutoff radius $r_{\text{cut}}$ while maintaining continuity. $\mathbf{r}_{ji} \in \mathbb{R}^3$ is the position of atom $j$ relative to atom $i$, and $r_{ji}$ is their distance. The coefficients $c_{nlm}^a$ express the discretization of the density on a basis of $n_{\max}$ radial functions $R_{nl}$ and spherical harmonics $Y_l^m$ that are the basic building blocks of the equivariant models described in Section 2. It should be noted that a different density $\rho_{i,a}(\mathbf{x})$ is

defined for each of the $a_{\max}$ chemical elements in the neighborhood, and that the sum only includes neighboring atoms whose chemical element $a_j$ matches $a$.

These densities can be used to define kernels that fulfill the equivariance condition (3)

$$\mathrm{k}_{\mu\mu'}^{\nu,\lambda}(A_i, A_{i'}') = \int \mathrm{d}\hat{R}\, D_{\mu\mu'}^{\lambda}(\hat{R}) \left( \sum_a \int \rho_{i,a}(\mathbf{x})\, \rho_{i',a}(\mathbf{R}^{-1}\mathbf{x})\, \mathrm{d}\mathbf{x} \right)^{\nu}, \qquad (5)$$

where $\nu$ will be referred to as the correlation order of the kernel, and the other symbols carry the same meaning as in (3). The $\nu = 2$ special case has been used to machine learn tensorial properties of atomistic systems in Grisafi et al. (2018). The kernels in (5) contain correlated information about at most $\nu$ neighbors in each atomic neighborhood ($A_i$ and $A_{i'}'$). This is because the density expansion in (4) is a simple sum over neighbors, and it is raised to the power of $\nu$, while all other operations (the inner integral and the rotation) are linear. As a result, these kernels are intrinsically body-ordered: $\mathrm{k}_{\mu\mu'}^{\nu,\lambda}$ can describe physical interactions up to body-order $\nu + 1$ (the center of the representation and $\nu$ neighbors), but not higher.

**Wigner kernels through Wigner iterations**. As detailed in Appendix C, symmetry-adapted kernels of the form given in (5) can be computed by first evaluating body-ordered equivariant *representations* (in the form of discretized correlations of the neighbor density (4)) and then computing their scalar products. These are the same representations that underlie MTP, ACE and NICE feature-space models, and that are very closely related to the representations that are implicitly generated by equivariant neural networks (Nigam et al., 2022a; Batatia et al., 2022a). Such a formulation highlights the positive-semi-definiteness of the kernels; however, performing such computations is impractical for $\nu > 2$: on one hand, kernel regression is then equivalent to linear regression on the starting features; on the other, the number of features one needs to compute to evaluate the kernel without approximations grows exponentially with $\nu$.

Our main result, which we will refer to as a Wigner iteration, is that high-$\nu$ kernels can be computed following an alternative route by combining lower-order kernels iteratively:

$$\mathrm{k}_{\mu\mu'}^{(\nu+1),\lambda}(A_i, A_{i'}') = \sum_{\substack{l_1 m_1 m_1' \\ l_2 m_2 m_2'}} \langle l_1 m_1; l_2 m_2 | \lambda\mu \rangle \, \mathrm{k}_{m_1 m_1'}^{\nu, l_1}(A_i, A_{i'}') \, \mathrm{k}_{m_2 m_2'}^{1, l_2}(A_i, A_{i'}') \, \langle l_1 m_1'; l_2 m_2' | \lambda\mu' \rangle, \quad (6)$$

where $\langle l_1 m_1; l_2 m_2 | \lambda\mu \rangle$ are Clebsch-Gordan coefficients. The proof (shown in Appendix B) follows from (5) and the relationships between Wigner D-matrices and Clebsch-Gordan coefficients. Although truncated in its angular parameters, this formulation of the high-order kernels is entirely lossless in terms of the radial basis and the dimension of composition (chemical element) space. Indeed, Appendix C shows how the Wigner kernel formulation corresponds to the infinite-width limit of equivariant feature-space linear models and neural networks.

In order to initialize the iterations in (6), only the $\nu = 1$ equivariant kernels $\mathrm{k}_{\mu\mu'}^{1,\lambda}$ are needed. Their expression, which follows immediately from (4) and (5), is given in Appendix D, along with details of its cheap evaluation. Equivariance with respect to inversion is discussed in Appendix E, and it results in the incorporation of a parity index $\sigma$, so that the full notation for an $O(3)$-equivariant kernel is $\mathrm{k}_{\mu\mu'}^{\nu,\lambda\sigma}$. Finally, we also define one-body, $\nu = 0$ kernels as $\mathrm{k}_{\mu\mu'}^{0,\lambda\sigma}(A_i, A_{i'}') = \delta_{\lambda 0}\delta_{\sigma 1}\delta_{a_i a_{i'}}$, which describe similarity of two environments based exclusively on the chemical elements of the central atoms $a_i$ and $a_{i'}$.

**Scaling and computational cost**. The calculation of the atom-centered density correlations that underlie linear and non-linear equivariant point cloud models entails an exponential scaling of the equivariant feature set size as a function of $\nu_{\max}$ (Nigam et al., 2020; Dusson et al., 2022), which is the consequence of a use of a radial-element basis of size $(a_{\max} n_{\max})$ out of which one effectively computes a sequence of outer products, affording a scaling of $\mathcal{O}((a_{\max} n_{\max})^{\nu})$. Computing Wigner kernels as scalar products of such equivariant features (see Appendix C) would present the same problems and require aggressive truncation of the basis. The calculation through a Wigner iteration can be understood as a tensor contraction strategy to compute the very same quantity, while avoiding the intermediate evaluation of these outer products (see the schematics in Figure 1), so that it is possible to use a converged basis while achieving a linear scaling with respect to $\nu_{\max}$. The scaling of the Wigner iteration with respect to its hyperparameters is discussed in Appendix F. Only the

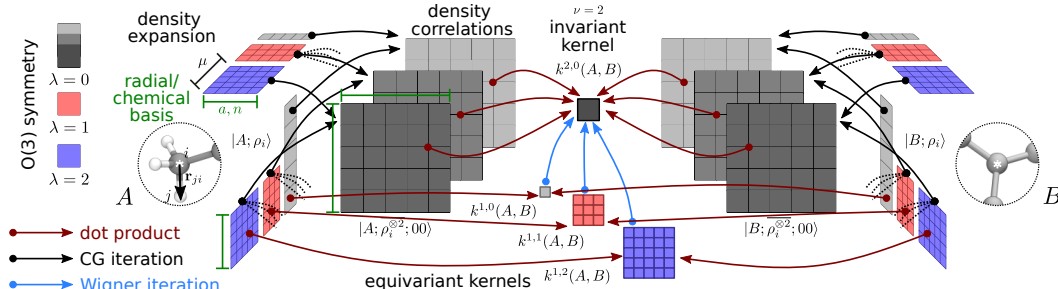

Figure 1: A schematic comparison of the calculation of body-ordered representations and kernels through Clebsh-Gordan products (black arrows) and through Wigner iterations (blue arrows), illustrated for the case of the $\nu = 2$ invariant kernel $k^{2,0}$. The former entails a scaling of representation size and computational cost with the square of the basis size, $(a_{\max}n_{\max})^2$. The latter computes immediately a kernel and is therefore independent on basis size. Further feature-space iterations increase exponentially the complexity, whereas kernel-space iterations are independent on the body order $\nu$.

angular basis has to be truncated at a maximum angular momentum order $\lambda_{\max}$, and the scaling is steeper ($\lambda_{\max}^7$) relative to traditional SO(3)-symmetrized products ($\lambda_{\max}^5$). Fortunately, as we shall see in Section 4.1, Wigner kernels exhibit excellent performance even with low $\lambda_{\max}$.

# 4 BENCHMARKS AND DISCUSSION

Having discussed the formulation and the theoretical scaling of Wigner kernels, we now proceed to assess their behavior in practical regression tasks, focusing on applications to atomistic machine learning. We refer the reader to Appendix G for a discussion of the implementation details, and to Appendix I for a list of the hyperparameters of the models. We consider four cases that allow us to showcase the accuracy of our framework: a system that is expected to exhibit strong many-body effects, one that requires high-resolution descriptors, and two classical benchmark datasets for organic molecules, including regression of a tensorial target.

## 4.1 ABLATION STUDIES: GOLD CLUSTER AND RANDOM METHANE DATASETS

In the first instance, we test the behavior of the Wigner kernels on two datasets that fully display the relative importance of its different body-ordered and angular components, respectively, while comparing the proposed model to its most closely related counterparts.

It is clear that the scaling properties of the Wigner kernel model discussed in Sec. 3 make it especially advantageous for systems requiring a high-body-order description of the potential energy surface. Metallic clusters often exhibit non-trivial finite-size effects due to the interplay between surface and bulk states (Li et al., 2013), and they have therefore been used in the past as prototypical benchmarks for many-body ML models (Zeni et al., 2018). As a particularly challenging test case, we consider a publicly-available dataset (Goldsmith & Ghiringhelli, 2016) of MD trajectories of gold clusters of different size (Goldsmith et al., 2019). From these trajectories, we select $105\,092$ uncorrelated structures for use in this work.

The need for high-body-order terms is clear when comparing results for models based on exponential WKs truncated at different orders of $\nu$ (Fig. 2). $\nu = 2$ and (to a lesser extent) $\nu = 3$ models result in saturating learning curves. A comparison with SOAP-based models reveals the likely source of the increased performance of the Wigner kernels. Indeed linear SOAP, which is a $\nu_{\max} = 2$ model, shows very similar performance to its $\nu = 2$ WK analogue. The same is true for squared-kernel SOAP-GPR, which closely resembles the learning curve of a Wigner kernel construction for which $\nu_{\max} = 2$ and the resulting kernels are squared - the difference probably due to the different functional form of the two kernels, and the presence of higher-$l$ components in the density for SOAP-GPR. A true $\nu_{\max} = 4$ kernel, that incorporates *all* five-body correlations, significantly outperforms both squared-kernel learning curves, demonstrating the advantages of explicit body-

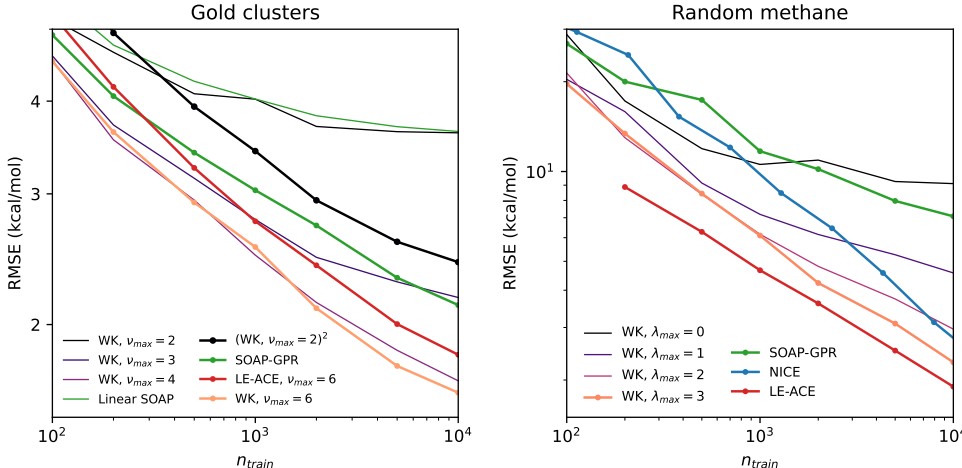

Figure 2: (left) Learning curves for the electronic free energy of gold clusters. Different curves correspond to invariant Wigner kernels of increasing body-order, as well as a construction where a linear combination of Wigner kernels up to $\nu_{\max} = 2$ is squared. A linear SOAP (Bartók et al., 2013) model, a SOAP-GPR (Deringer et al., 2021) model built with a squared kernel, and a LE-ACE (Bigi et al., 2022) model are also shown. The hyperparameters for all models are discussed in Appendix I. (right) Learning curves for the energy of random $CH_4$ configurations, comparing different models. The LE-ACE and NICE curves are from Refs. Bigi et al. (2022) and Nigam et al. (2020), respectively. Hyperparameters for all other models are discussed in Appendix I. All results are the average of 10 random train/test splits within the respective datasets, where the number of test structures is kept constant at 1 000. Figures with error bars relative to these random splits are provided in Appendix K. We note that REANN (Zhang et al., 2022) and PET (Pozdnyakov & Ceriotti, 2023) achieved higher accuracy on this dataset by also learning from forces.

ordering. We conclude with a comparison between the $\nu_{\max} = 6$ WKs and $\nu_{\max} = 6$ Laplacian-Eigenbasis (LE) ACE models. For the latter, we used the same radial transform presented in (Bigi et al., 2022), and we optimized its single hyperparameter. Although it might be possible to further tune the performance of LE-ACE by changing the functional form of the radial transform altogether, the comparison with the Wigner kernel learning curve suggests that the kernel-space basis employed in the Wigner kernels might be advantageous in geometrically inhomogeneous datasets such as this.

As a second example, we test the Wigner kernels on a random gas-phase $CH_4$ dataset (Pozdnyakov et al., 2020b;a) which we expect to be very challenging for the proposed Wigner kernel model, as it is intrinsically limited in body-order and almost random in its configurations, so that using a training-set kernel basis provides close to no advantages. More importantly, this dataset requires very careful convergence of the angular basis (Nigam et al., 2022a; Bigi et al., 2022), which is problematic in view of the steep $\lambda_{\max}$ scaling of Wigner iterations.

With all these potential problems, Wigner kernels achieve a remarkable level of accuracy, outperforming SOAP-GPR and NICE, and being competitive with LE-ACE despite using only $\lambda_{\max} = 3$. Similar low-$\lambda_{\max}$ effects have been noticed in many recent efforts to machine-learn interatomic potentials (Batzner et al., 2022b; Batatia et al., 2022a; Musaelian et al., 2023; Batatia et al., 2022c). By providing a functional form that spans the full space of density correlations at a given level of angular truncation, Wigner kernels can help rationalize why low-$\lambda_{\max}$ models can perform well. Indeed, due to the form of the Wigner iterations, $k^{(\nu)}$ does not report *exclusively* on $(\nu + 1)$-body correlations, but also on all lower-order ones, and the tensor-product form of the kernel space incorporates higher frequency components in their functional form, much like $\sin^2(\omega x)$ contains components with frequency $2\omega$. We investigate and confirm this hypothesis in Appendix J by decomposing the angular dependence of high-$\nu$ kernels into their frequency components. This explains why aggessively truncated equivariant ML models (Schütt et al., 2021; Batzner et al., 2022a) can achieve high accuracy in the prediction of interatomic potentials.

Table 1: Performance comparison of the Wigner kernel model with the best literature models on the rMD17 dataset (Christensen & Von Lilienfeld, 2020) in its smaller version (50 randomly selected training structures). Accuracies of energies (E) and forces (F) are given as MAE in eV and eV/Å, respectively. The best results on each target are highlighted in bold. The timings refer to the average of one energy + force prediction over all molecules in the dataset, on a Nvidia V100 GPU in double precision. It should be noted that MACE was found to be faster than NequIP (Batatia et al., 2022b) and that no GPU implementation of LE-ACE is available at the moment.

| Molecule | LE-ACE | | NequIP | | MACE | | WK | |
|---|---|---|---|---|---|---|---|---|
| | E | F | E | F | E | F | E | F |
| Aspirin | 22.4 | 59.1 | 19.5 | 52.0 | **17.0** | **43.9** | **17.0** | 50.2 |
| Azobenzene | 9.9 | 27.5 | 6.0 | 20.0 | **5.4** | **17.7** | 7.9 | 25.6 |
| Benzene | 0.135 | 1.44 | 0.6 | 2.9 | 0.7 | 2.7 | **0.131** | **1.31** |
| Ethanol | 6.6 | 32.0 | 8.7 | 40.3 | 6.7 | 32.6 | **5.9** | **30.8** |
| Malonaldehyde | 11.3 | 50.9 | 12.7 | 52.5 | 10.0 | **43.3** | **8.9** | 43.8 |
| Naphthalene | 2.9 | 13.9 | **2.1** | 10.0 | **2.1** | **9.2** | 2.5 | 12.5 |
| Paracetamol | 14.3 | 45.1 | 14.3 | 39.7 | **9.7** | **31.5** | 10.2 | 37.2 |
| Salicylic acid | 8.3 | 36.7 | 8.0 | 35.0 | **6.5** | **28.4** | 6.8 | 31.9 |
| Toluene | 4.1 | 18.4 | 3.3 | 15.1 | **3.1** | **12.1** | 3.4 | 16.4 |
| Uracil | 5.7 | 30.7 | 7.3 | 40.1 | **4.4** | **25.9** | 5.1 | 27.8 |
| Average latency | - | | - | | 92 ms | | 56 ms | |

## 4.2 RMD17 DATASET

We proceed our investigation on the rMD17 dataset (Christensen & Von Lilienfeld, 2020), which assesses the accuracy achieved by models when learning potential energy surfaces of small organic molecules. When using the derivative learning scheme in Chmiela et al. (2017b), Wigner kernels are shown to systematically outcompete the ACE implementation that performs best on this benchmark (LE-ACE, Bigi et al. (2022)), showing the advantages of a fully converged radial-chemical description. The proposed model is also competitive in accuracy with equivariant neural networks such as NequIP (Batzner et al., 2022b) and MACE (Batatia et al., 2022b), while operating at a reduced computational cost. Using atomic neighborhoods as support points rather than full structures would further reduce the cost of the Wigner kernels by roughly an order of magnitude (by eliminating the sum over $i'$ in (1)) while causing little to no deterioration in accuracy. Exploiting the sparsity of the Clebsch-Gordan matrices, as is done in MACE, would also improve the efficiency of the proposed model. Finally, it is also worth mentioning that, similar to ACE and Allegro (Musaelian et al., 2023), but unlike NequIP and MACE, Wigner kernels are entirely local, as they do not incorporate message-passing operations. This would greatly simplify the parallelization of inference for large-scale calculations.

## 4.3 QM9 DATASET

Wigner kernels avoid the unfavorable scaling of traditional body-ordered models with respect to the number of chemical elements in the system. This property is particularly useful when dealing with chemically diverse datasets. An example is that of the popular QM9 dataset (Ramakrishnan et al., 2014), which contains 5 elements (H, C, N, O, F).

We build Wigner kernel models for two atomic-scale properties within this dataset, and, to illustrate the transferability of our model, we use the same hyperparameters for both fits (see Appendix I).

**Molecular dipoles.** We begin the investigation with a covariant learning exercise. This consists of learning the dipole moment vectors $\boldsymbol{\mu}$ of the molecules in the QM9 dataset (Veit et al., 2020). In the small-data regime, Wigner kernels have a similar performance to that obtained by optimized $\lambda$-SOAP kernels in Veit et al. (2020), but they completely avoid the saturation for larger train set size (Fig. 3). The improved performance of the Wigner kernels is a clear indication of the higher descriptive power that is afforded by the use of a full body-ordered equivariant kernel, as opposed to

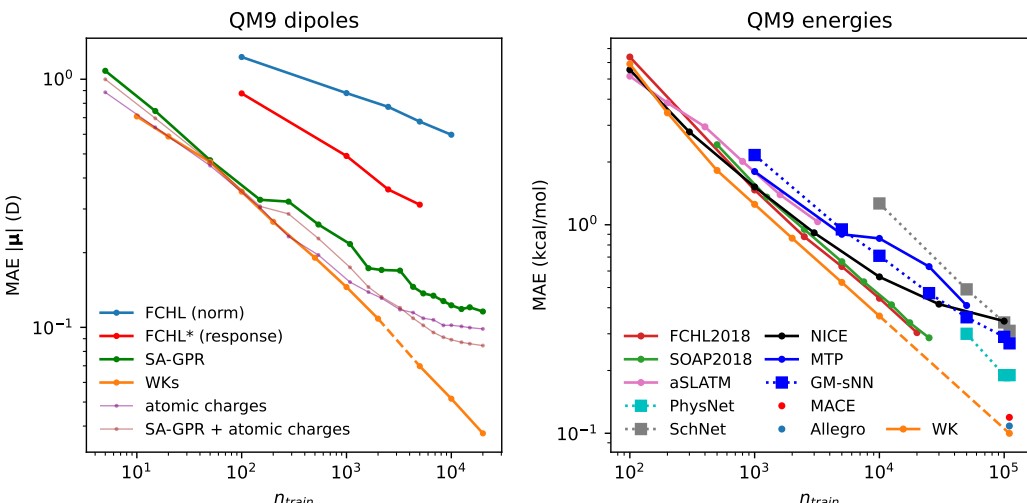

Figure 3: (left) Learning curves for the prediction of molecular dipole moments in the QM9 datasets. Different curves correspond to FCHL kernels (Faber et al., 2018), the dipole models presented in Veit et al. (2020), and Wigner kernels. It should be noted how the models that use atomic charges can account for the macroscopic component of the dipole moment that arises due to charge separation, while the others can only predict dipole moments as a sum of local atom-centered contributions. The dashed line in the WK learning curve represents a change in the fitting procedure: the points before the dashed line are obtained as highlighted in Appendix G, while the points after the dashed line are obtained with the same cross-validation procedure, but using a less expensive 2-dimensional grid search over the kernel mixing parameters (Appendix G.2). The accuracy of the model does not seem to be affected by this change. (right) Selection of the best QM9 literature models for which learning curves are available: FCHL (Faber et al., 2018), SOAP-GPR (Willatt et al., 2018), aSLATM (Huang & von Lilienfeld, 2016), PhysNet (Unke & Meuwly, 2019), SchNet (Schütt et al., 2018), NICE (Nigam et al., 2020), MTP (Shapeev, 2016), GM-sNN (Zaverkin & Kästner, 2020). A few selected neural networks whose learning curves are not available are also shown on the *full* QM9 dataset (right-most isolated points); more are available in Table 2. The dashed line in the WK learning curve represents a change in the fitting procedure. The points to its left are obtained by averaging 10 runs with random train/test splits, where 1 000 test structures are employed, and cross-validation is conducted within the training set as described in Appendix G. Instead, for consistency with the literature models trained on the full QM9 dataset (Table 2), the last point is averaged over 16 random train/validation/test splits where validation is conducted on a dedicated validation set via a grid search, as discussed in the caption of Table 2. Figures with error bars relative to these random splits are provided in Appendix K.

the combination of linear covariant $\nu = 2$ kernels and non-linear scalar kernel that is used in current applications of SA-GPR.

**Energies**. Finally, we test the Wigner kernel model on the ground-state energies of the QM9 dataset. The corresponding learning curves are shown in Fig. 3. Wigner kernels significantly improve on other kernel methods such as SOAP and FCHL in the low-data regime. As in the case of $CH_4$, the WK model is truncated at a low angular threshold ($\lambda_{max} = 3$). However, the corresponding learning curve shows no signs of saturation, possibly for the same reasons we highlighted in Sec. 4.1. Similarly, a relatively low maximum body-order ($\nu_{max} = 4$) does not seem to impact the accuracy of the model, most likely because stable organic molecules have, with few exceptions, atoms with only up to four nearest neighbors. On the full QM9 dataset, Wigner kernels also achieve state-of-the-art accuracy, as shown in the last point of the WK learning curve and in Table 2. The impressive performance of the Wigner kernels on this exercise shows the suitability of the proposed model to, for instance, screening of pharmaceutical targets or prediction of chemical shifts from single equilibrium configurations. This stands in contrast to the other datasets we have investigated, which are better suited to assess the quality of a model in approximating a property surface for atomistic simulations.

Table 2: Performance comparison of the Wigner kernel model with a selection of the best literature models on the full QM9 dataset, as presented in Musaelian et al. (2023). The WK values are the mean and standard deviation of 16 runs on different random train/validation/test splits. In particular, the training set contains $110\,000$ random structures, the validation set another $10\,000$, and all the remaining QM9 structures constitute the test set, for consistency with Musaelian et al. (2023).

| Model | $U_0$ | $U$ | $H$ | $G$ | Avg. |
|---|---|---|---|---|---|
| NoisyNodes (Godwin et al., 2021) | 7.3 | 7.6 | 7.4 | 8.3 | 7.65 |
| SphereNet (Liu et al., 2021) | 6.3 | 6.4 | 6.3 | 7.8 | 6.70 |
| DimeNet++ (Klicpera et al., 2020) | 6.3 | 6.3 | 6.5 | 7.6 | 6.68 |
| ET (Thölke & De Fabritiis, 2022) | 6.2 | 6.4 | 6.2 | 7.6 | 6.60 |
| PaiNN (Schütt et al., 2021) | 5.9 | 5.8 | 6.0 | 7.4 | 6.28 |
| MACE (Kovacs et al., 2023) | 5.2 (0.2) | **4.1** | 4.7 | **5.5** | 4.88 |
| Allegro (Musaelian et al., 2023) | 4.7 (0.2) | 4.4 | 4.4 | 5.7 | 4.80 |
| TensorNet (Simeon & De Fabritiis, 2023) | **4.3** (0.3) | 4.3 (0.1) | 4.3 (0.2) | 6.0 (0.1) | 4.72 |
| Wigner Kernels | **4.3** (0.1) | 4.2 (0.2) | **4.2** (0.2) | 6.0 (0.1) | **4.68** |

## 5 CONCLUSIONS

In this work, we have presented the Wigner iteration as a practical tool to construct rotationally equivariant "Wigner kernels" for use in symmetry-adapted Gaussian process regression on 3D point clouds. We have then applied them to machine learn the atomistic properties of molecules and clusters. The proposed kernels are explicitly body-ordered – i.e. they provide explicit universal approximation capabilities (Dusson et al., 2022) for properties that depend simultaneously on the correlations between the positions of $\nu + 1$ points – and can be thought as the kernels corresponding to the infinite-width limit of several families of body-ordered models. This extends the well-known equivialence between infinitely wide neural networks and Gaussian processes (Neal, 1996; Williams, 1996; Lee et al., 2017) from a statistical context to the one of geometric representations. Whereas the full feature-space evaluation of body-ordered models leads to an exponential increase of the cost with $\nu$, a kernel-space evaluation is naturally adapted to the training structures, and it avoids the explosion in the number of equivariant features that arises from the use of an explicit radial-chemical basis. The scaling properties of the Wigner iterations make the new model particularly suitable for datasets which are chemically diverse, which are expected to contain strong high-body-order effects, and/or which involve a very inhomogeneous distribution of molecular geometries.

Our benchmarks demonstrate the excellent performance of KRR models based on Wigner iterations on a variety of different atomistic problems. The ablation studies on gold clusters and gas-phase methane molecules fully reveal the strengths and weaknesses of the proposed model. In particular, the results for a random $CH_4$ dataset suggest that Wigner kernels incorporate high-resolution basis functions even when they are built with a moderate angular momentum threshold, which is reassuring given the steep scaling of the computational cost with $\lambda_{\mathrm{max}}$. The chemically diverse rMD17 and QM9 datasets allow us to showcase the state-of-the-art performance of the proposed model when learning energies, forces, and vectorial dipole moments. The fact that a kernel model can match the performance of extensively tuned equivariant neural networks testifies to the importance of understanding the connection between body-ordered correlations, the choice and truncation of a feature-space basis, and the introduction of scalar non-linearities in equivariant models.

Besides this fundamental role to test the complete-basis limit of density-correlation models, it is clear that Wigner iterations can be incorporated into practical applications. Our model achieves high efficiency on small molecules, and using a sparse kernel formalism will allow to further reduce its computational cost and apply the model to much larger systems. Finally, the Wigner iteration could also be applied outside a pure kernel regression framework: from the calculation of non-linear equivariant functions, to the use in Gaussian process classifiers (Rasmussen, 2006), to the inclusion as a layer in an equivariant architecture, the ideas we present here open up an original research direction in the construction of symmetry-adapted, physically inspired models for chemistry, materials science, and more in general any application whose inputs can be conveniently described in terms of a 3D point cloud.

## 6 REPRODUCIBILITY STATEMENT

The code used to generate the Wigner kernel results is available as Supplementary Material, along with instructions on how to use it. Hyperparameters for all the numerical experiments are given in Appendix I.

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

## A    TABLE OF SYMBOLS

| Symbol | Description |
| --- | --- |
| $A, A', B$ | Atomic-scale structures. Effectively a shorthand for the positions and chemical elements of the atoms within the structures |
| $i, i'$ | Individual atoms within an atomic-scale structure |
| $A_i, A'_{i'}$ | Spherical neighborhoods of atoms $i$ and $i'$ within structures $A$ and $A'$ |
| $\lambda, l_1, l_2$ | Degree of spherical harmonics, labels of irreps of the $SO(3)$ group |
| $\mu, m_1, m_2, m'_1, m'_2$ | Order of spherical harmonics, index of basis functions of irreps of $SO(3)$ corresponding to $\lambda, l_1, l_2, l_1, l_2$, respectively |
| $y_\lambda^\mu(A)$ | Equivariant property of structure $A$ which transforms as a spherical harmonic of degree $\lambda$ and order $\mu$ |
| $\tilde{y}_\lambda^\mu(B)$ | Prediction of an equivariant property of structure $B$ which transforms as a spherical harmonic of degree $\lambda$ and order $\mu$ |
| $\hat{R}, \hat{R}'$ | 3D rotation operators |
| $\mathbf{D}^\lambda(\hat{R}), D_{\mu\mu'}^\lambda(\hat{R})$ | Wigner D-matrix of degree $\lambda$, and its matrix elements, corresponding to $\hat{R}$ |
| $\mathbf{R}$ | Cartesian rotation matrix associated to $\hat{R}$ |
| $k_{\mu\mu'}^\lambda$ | A generic equivariant kernel that satisfies (3) |
| $a$ | A chemical element |
| $a_i$ | The chemical element of atom $i$ |
| $\mathbf{x}$ | A 3D coordinate |
| $\rho_{i,a}(\mathbf{x})$ | Atomic density of element $a$ in the neighborhood of atom $i$ |
| $\mathbf{r}_{ji}$ | 3D vector from atom $i$ to atom $j$ |
| $r_{ji}$ | Distance from atom $i$ to atom $j$ |
| $g(\mathbf{x} - \mathbf{r}_{ji})$ | A 3D Gaussian function centered at $\mathbf{r}_{ji}$ |
| $f_{cut}(r_{ji})$ | Cutoff function evaluated at $r_{ji}$ |
| $\nu$ | A correlation order, or, equivalently, a body order minus one |
| $k_{\mu\mu'}^{\nu,\lambda}$ | An equivariant kernel that satisfies (3) and has correlation order $\nu$ |
| $\sigma$ | Label for inversion symmetry: +1 is symmetric, -1 is antisymmetric |
| $\lambda_{\max}$ | Hyperparameter: the maximum value of $\lambda$ |
| $\nu_{\max}$ | Hyperparameter: the maximum value of $\nu$ |
| $n_{\max}$ | Number of radial functions in the discretization of $\rho_i$ |
| $a_{\max}$ | Number of chemical elements in the dataset |

## B DERIVATION OF THE WIGNER ITERATION

In this Appendix, we derive Eq. (6). In order to make the notation more compact, we define unsymmetrized versions of the equivariant kernels in (5) which only contain the inner integral:

$$\mathrm{k}^{\nu}(A_i, \hat{R}A_i') = \left( \int \rho_i(\mathbf{x})\, \rho_{i'}(\mathbf{R}^{-1}\mathbf{x})\, \mathrm{d}\mathbf{x} \right)^{\nu}, \tag{7}$$

from which it is straightforward to see that

$$\mathrm{k}^{\nu+\nu'}(A_i, \hat{R}A_i') = \mathrm{k}^{\nu}(A_i, \hat{R}A_i')\, \mathrm{k}^{\nu'}(A_i, \hat{R}A_i'). \tag{8}$$

According to its behavior upon the relative rotation of its two densities $\hat{R}$, the $\mathrm{k}^{\nu}$ kernel can be decomposed into $\mathrm{k}^{\nu,\lambda}_{\mu\mu'}$ contributions. Within the space of rotations $\hat{R}$, the latter kernels are effectively the expansion coefficients of $\mathrm{k}^{\nu}$ in the basis of the Wigner D-matrices $D^{\lambda}_{\mu\mu'}(\hat{R})$, so that

$$\mathrm{k}^{\nu}(A_i, \hat{R}A_{i'}) = \sum_{\lambda\mu\mu'} \mathrm{k}^{\nu,\lambda}_{\mu\mu'}(A_i, A_{i'})\, D^{\lambda}_{\mu\mu'}(\hat{R}) \tag{9}$$

and

$$\mathrm{k}^{\nu,\lambda}_{\mu\mu'}(A_i, A_{i'}) = \int D^{\lambda}_{\mu\mu'}(\hat{R})^*\, \mathrm{k}^{\nu}(A_i, \hat{R}A_{i'})\, d\hat{R}, \tag{10}$$

which corresponds to (5) in the main text.

A simple combination of Eqs. 8, 9 and 10 leads to our main result:

$$\mathrm{k}^{\nu+1,\lambda}_{\mu\mu'}(A_i, A_{i'}) \overset{(10)}{=} \int D^{\lambda}_{\mu\mu'}(\hat{R})^*\, \mathrm{k}^{\nu+1}(A_i, \hat{R}A_{i'})\, \mathrm{d}\hat{R} \overset{(8)}{=}$$

$$\int D^{\lambda}_{\mu\mu'}(\hat{R})^*\, \mathrm{k}^{\nu}(A_i, \hat{R}A_{i'})\, \mathrm{k}^{1}(A_i, \hat{R}A_{i'})\, d\hat{R} \overset{(9)}{=}$$

$$\sum_{\substack{l_1 m_1 m_1' \\ l_2 m_2 m_2'}} \mathrm{k}^{\nu,l_1}_{m_1 m_1'}(A_i, A_{i'})\, \mathrm{k}^{1,l_2}_{m_2 m_2'}(A_i, A_{i'})$$

$$\int D^{\lambda}_{\mu\mu'}(\hat{R})^*\, D^{l_1}_{m_1 m_1'}(\hat{R})\, D^{l_2}_{m_2 m_2'}(\hat{R})\, d\hat{R} =$$

$$\sum_{\substack{l_1 m_1 m_1' \\ l_2 m_2 m_2'}} C^{l_1 l_2 \lambda}_{m_1 m_2 \mu}\, \mathrm{k}^{\nu,l_1}_{m_1 m_1'}(A_i, A_{i'})\, \mathrm{k}^{1,l_2}_{m_2 m_2'}(A_i, A_{i'})\, C^{l_1 l_2 \lambda}_{m_1' m_2' \mu'}, \tag{11}$$

where, in the last equality, we have used a well-known property of the Wigner D-matrices which relates them to the Clebsch-Gordan coefficients $C^{l_1 l_2 L}_{m_1 m_2 M}$.

## C DENSITY-CORRELATION VIEW OF THE WIGNER ITERATION

An alternative derivation of the Wigner iteration (Eq. 5) reveals a direct connection with frameworks that operate in feature space (ACE (Drautz, 2019), NICE (Nigam et al., 2020)). To achieve this, we will use the notation from Nigam et al. (2020), in which the expansion coefficients of the neighbor density are written as

$$\langle an|\rho_i^{\otimes 1}; lm\rangle = c^a_{nlm}(A_i) = \int \mathrm{d}\mathbf{x}\, \rho_{i,a}(\mathbf{x}) R_{nl}(x) Y_l^m(\hat{\mathbf{x}}).$$

Disregarding here and in the following the elemental index (for all practical purposes one can consider $n$ as a compound index enumerating the radial-element basis), one can then express high order density correlations iteratively, based on a Clebsch-Gordan SO(3) product

$$\langle q|\rho_i^{\otimes(\nu+1)}; \lambda\mu\rangle = \sum_{m_1 m_2} \langle q'|\rho_i^{\otimes\nu}; l_1 m_1\rangle\, \langle n|\rho_i^{\otimes 1}; l_2 m_2\rangle\, \langle l_1 m_1; l_2 m_2|\lambda\mu\rangle.$$

Here $q = \{q, n, l_1, l_2\}$ is a compound index that spans the tensor product space of the $\nu$-order correlations, the radial-elemental basis, and the angular momenta of the two factors. This Clebsch-Gordan product is also the central operation in equivariant networks Anderson et al. (2019); Batzner et al. (2022b), in which it is combined with contraction operations to avoid the increase in feature size. Having recalled these results, which also clearly explain why frameworks based on the equivariant iteration (C) tend to lead to exponential increase of the dimensionality of feature space, we can proceed to an alternative derivation of the Wigner iteration that shows how it effectively avoids the explicit evaluation of the tensor product of features by changing the summation order in the scalar-product definition of Wigner kernels. Starting from (5), we obtain

$$
\begin{aligned}
\mathrm{k}_{\mu\mu'}^{\nu,\lambda}(A_i, A_{i'}) = \int \mathrm{d}\hat{R}\, D_{\mu\mu'}^{\lambda}(\hat{R})^* \int \rho_i(\mathbf{x}^\nu)\, \rho_{i'}((\mathbf{R}^{-1}\mathbf{x})^\nu)\, \mathrm{d}\mathbf{x}^\nu = \\
\int \mathrm{d}\hat{R}\, D_{\mu\mu'}^{\lambda}(\hat{R})^* \sum_{LMq} \langle \rho_i^{\otimes\nu}; LM|q\rangle \langle q|\hat{R}\rho_{i'}^{\otimes\nu}; LM\rangle = \\
\int \mathrm{d}\hat{R}\, D_{\mu\mu'}^{\lambda}(\hat{R})^* \sum_{LMq} \langle \rho_i^{\otimes\nu}; LM|q\rangle \sum_{M'} D_{MM'}^{L}(\hat{R}) \langle q|\rho_{i'}^{\otimes\nu}; LM'\rangle = \\
\delta_{L\lambda}\delta_{M\mu}\delta_{M'\mu'} \sum_{LMq} \langle \rho_i^{\otimes\nu}; LM|q\rangle \sum_{M'} \langle q|\rho_{i'}^{\otimes\nu}; LM'\rangle = \\
\sum_q \langle \rho_i^{\otimes\nu}; \lambda\mu|q\rangle \langle q|\rho_{i'}^{\otimes\nu}; \lambda\mu'\rangle. \quad (12)
\end{aligned}
$$

The first equality is a rearranged version of (5), the second is a change of basis from real space to a rotationally symmetrized $LMq$ basis, the third and fourth employ properties of the Wigner D-matrices, and the fifth follows immediately. This equation shows that Wigner kernels correspond to scalar products between body-ordered density-correlation features, computed in the limit of a complete basis set expansion for the neighbor density.

Now, using the iterative equivariant construction presented in Nigam et al. (2020), it can be noted that

$$
\begin{aligned}
\mathrm{k}_{\mu\mu'}^{\nu+1,\lambda}(A_i, A_{i'}) = \sum_q \langle \rho_i^{\otimes(\nu+1)}; \lambda\mu|q\rangle \langle q|\rho_{i'}^{\otimes(\nu+1)}; \lambda\mu'\rangle = \\
\sum_{\substack{l_1 m_1 m_1' \\ l_2 m_2 m_2' \\ q'n}} C_{m_1 m_2 \mu}^{l_1 l_2 \lambda} C_{m_1' m_2' \mu'}^{l_1 l_2 \lambda} \\
\langle \rho_i^{\otimes\nu}; l_1 m_1|q'\rangle \langle \rho_i^{\otimes 1}; l_2 m_2|n\rangle \langle n|\rho_{i'}^{\otimes 1}; l_2 m_2'\rangle \langle q'|\rho_{i'}^{\otimes\nu}; l_1 m_1'\rangle = \\
\sum_{\substack{l_1 m_1 m_1' \\ l_2 m_2 m_2'}} C_{m_1 m_2 \mu}^{l_1 l_2 \lambda}\, \mathrm{k}_{m_1 m_1'}^{\nu,l_1}(A_i, A_{i'})\, \mathrm{k}_{m_2 m_2'}^{1,l_2}(A_i, A_{i'}) C_{m_1' m_2' \mu'}^{l_1 l_2 \lambda}, \quad (13)
\end{aligned}
$$

which is an alternative derivation of Eq. 6). As a final note, we use Eq. (12) to calculate the $\mathrm{k}_{\mu\mu'}^{(1)\lambda}$ kernels (Appendix D). This is very convenient, as most atomistic machine learning software packages can calculate features of the $\langle n|\rho_{i'}^{\otimes 1}; \lambda\mu\rangle$ kind, otherwise known as density expansion coefficients (Musil et al., 2021).

## D  First-order Wigner kernels

The $\nu = 1$ Wigner kernels are relatively simple to define as a double sum over neighbors using (4) and (5):

$$
\mathrm{k}_{\mu\mu'}^{1,\lambda}(A_i, A_{i'}') = \delta_{a_i a_{i'}} \int \mathrm{d}\hat{R}\, D_{\mu\mu'}^{\lambda}(\hat{R}) \sum_{\substack{j \in A_i \\ j' \in A_{i'}'}} \delta_{a_j a_{j'}} f_{\mathrm{cut}}(r_{ji})\, f_{\mathrm{cut}}(r_{j'i'}) \\
\times \int g(\mathbf{x} - \mathbf{r}_{ji})\, g(\mathbf{x} - \mathbf{R}^{-1}\mathbf{r}_{j'i'})\, \mathrm{d}\mathbf{x}, \quad (14)
$$

where the $\delta_{a_i a_{i'}}$ term simply indicates that kernels between atoms of different chemical species are set to zero.

The integrals in (14) could be evaluated analytically (Bartók et al., 2013), although in practice we compute them numerically as scalar products of an atom-centered density expansion, as explained in Appendix C. This is done for simplicity and to avoid quadratic scaling in the number of neighbors. See Appendix C for more information.

## E    EQUIVARIANCE WITH RESPECT TO INVERSION

In order to take into account equivariance with respect to inversion, we include a further index $\sigma$: $k_{\mu\mu'}^{\nu,\lambda\sigma}$. For more details, we redirect the reader to Nigam et al. (2020) and its Supplemental Information. Here, it suffices to say that the meaning of the $\sigma$ index with regards to inversion is similar to that of $\lambda$, $\mu$, and $\mu'$ with regards to rotations, and that Eq. 6 needs to be slightly modified as follows:

$$k_{\mu\mu'}^{\nu+1,\lambda\sigma}(A_i, A_{i'}) = \sum_{\substack{l_1 m_1 m_1' l_2 m_2 m_2' \\ \sigma=s_1 s_2 (-1)^{l_1+l_2+\lambda}}} k_{m_1 m_1'}^{\nu,l_1 s_1}(A_i, A_{i'}) \, k_{m_2 m_2'}^{1,l_2 s_2}(A_i, A_{i'}) \, C_{m_1 m_2 \mu}^{l_1 l_2 \lambda} \, C_{m_1' m_2' \mu'}^{l_1 l_2 \lambda}. \quad (15)$$

Finally, we note that the $\nu = 1$ kernels in Appendix D always have $\sigma = 1$ (Musil et al., 2021).

## F    SCALING ANALYSIS

Since the generation of $\nu = 1$ kernels (Appendix D) is computationally cheap, we will focus on the cost of the Wigner iterations. A simple inspection of Eq. 6 yields the scaling of the computational cost of the algorithm with respect to various convergence parameters.

$n_{\max}$ **and** $a_{\max}$**:**    Since the $n$ index only appears in the initial (and inexpensive) generation of $\nu = 1$ kernels, there is approximately no scaling associated with $n_{\max}$. The same is true for the element indices $a$, hence the cost of evaluating Wigner kernels is independent of the total number of elements in the system $a_{\max}$. If one wanted to avoid the dependence entirely, it would suffice to compute explicitly the double sum in Appendix D, which scales with the product of the number of neighbors in the two environments but does not include any basis.

$\lambda_{\max}$**:**    There are nine angular indices in (6): $\lambda$, $\mu$, $\mu'$, $l_1$, $m_1$, $m_1'$, $l_2$, $m_2$, and $m_2'$. While in principle they all scale linearly with $\lambda_{\max}$, $m_2$ and $m_2'$ are redundant due to the properties of the Clebsch-Gordan coefficients. Therefore, the model scales as $\lambda_{\max}^7$.

$\nu_{\max}$**:**    supposing the model is truncated at a correlation order of $\nu_{\max}$ (which will generate a physical model with a maximum body-order of $\nu_{\max} + 1$), $\nu_{\max} - 1$ Wigner iterations (Eq. 6) are needed. If all iterations are truncated at $\lambda \leq \lambda_{\max}$, the cost of the iterations is identical for any $\nu$, and therefore the cost of computing *all* terms scales linearly with $\nu_{\max}$. Note that if a single $\nu$ is desired, the cost can be lowered by performing Wigner iterations between kernels with $\nu > 1$, see Appendix H. More generally, the iteration (Eq. 6) can be trivially generalized to combine arbitrary equivariant kernels, which makes it possible to compute certain forms of symmetrized non-linear equivariant functions more efficiently.

$n_{\text{train}}$**.**    In a naive implementation of kernel methods, such as what we will use here for simplicity, training requires computing a $n_{\text{train}} \times n_{\text{train}}$ kernel matrix $\mathbf{K}$ and inverting it. Given the substantial cost of computing Wigner kernel entries, in almost all cases we consider the cost is dominated by the quadratic scaling in computing the kernels, and not by the cubic cost of the inversion. Inference has a cost scaling linearly with $n_{\text{train}}$. Most practical KRR implementations use low-rank approximations of the kernel matrix (as in the projected process approximation (Rasmussen, 2006)) that make the construction of the training matrix formally linear in $n_{\text{train}}$, and inference independent of it – similar to methods based on linear regression such as ACE or MTP. In practice, however, increasing the sparse set (or feature set) size is inevitable in order to avoid saturation of the accuracy as $n_{\text{train}}$ increases (Bigi et al., 2022).

## G  PRACTICAL IMPLEMENTATION

KRR models such as those discussed in Sec. 2.1 are fully defined by the choice of the kernel function, which therefore strongly affects the accuracy they can achieve. In our construction, there are two main steps which influence the final kernel function: the definition of the atomic densities and the mixing of different body-ordered kernels.

### G.1  DENSITY EXPANSION FORM

The general form of the density expansion has already been introduced in Sec. 2.2. Contrary to common practice, we allow the width of the Gaussians in (4) to vary with the distance from the central atom. In particular, we use L1-normalized Gaussians where the width increases exponentially with the distance,

$$g(\mathbf{x} - \mathbf{r}_{ji}) \sim \exp\left[-(\mathbf{x} - \mathbf{r}_{ji})^2/2(Ce^{r_{ji}/r_0})^2\right]. \tag{16}$$

The cutoff function is set to

$$f_{\text{cut}}(r_{ji}) = e^{-r_{ji}/r_0}, \tag{17}$$

and the density contributions are further multiplied by a shifted cosine function in the last $0.5$ Å before $r_{\text{cut}}$ to ensure that the predictions of the model (as well as their first derivatives) are smooth as neighbors enter and leave the atomic environments defined by the cutoff radius. This choice of Gaussian smearing and cutoff function assumes the exponential decrease of the magnitude and resolution of physical interactions with distance. The hyperparameters $C$ and $r_0$, that determine the the maximum resolution and the decay length, are optimized separately for each dataset via a grid search.

### G.2  EXPONENTIAL KERNELS

Similar to how a modulation of the neighbor density can be used to exploit the physical prior that interactions between atoms decay with distance, one can incorporate the common wisdom that low-order correlations dominate the contributions to atom-centered properties by building a the overall kernel as a linear combination of the body-ordered Wigner kernels:

$$k_{\mu\mu'}^{\lambda\sigma} = \sum_{\nu=0}^{\nu_{\max}} c_\nu \, k_{\mu\mu'}^{\nu,\lambda\sigma} \,. \tag{18}$$

While in principle all the $c_\nu$ could vary independently, we found an exponential-like parametrization of Eq. (18) to be particularly effective:

$$k_{\mu\mu'}^{\lambda\sigma} = c_0 \, k_{\mu\mu'}^{0,\lambda\sigma} + a \sum_{\nu=1}^{\nu_{\max}} \frac{b^\nu}{\nu!} k_{\mu\mu'}^{\nu,\lambda\sigma}, \tag{19}$$

A similar kernel construction was proposed in (Glielmo et al., 2018) for body-ordered invariant kenels, noting however that no practical algorithm existed for its evaluation.

In this context, the Wigner iteration provides an efficient way to evaluate a truncated Taylor expansion of the exponential by pre-computing the body-ordered kernels $k_{\mu\mu'}^{\nu,\lambda\sigma}$ up to $\nu = \nu_{\max}$. Furthermore, given that it can be applied to combine any pair of equivariant kernels, it would also allow to implement other, more efficient algorithms to evaluate an exponential (Moler & Van Loan, 2003), such as scaling and squaring (see Appendix H). In practice, however, we found very high-body-order interactions to be of marginal importance in our tests, so we prefer to evaluate the summation explicitly, as this also simplifies the optimization of the related hyperparameters. These are $c_0$, $a$, and $b$, and they are optimized by dual annealing using 10-fold cross-validation within the training set. Given that these three coefficients also set the overall scale of the kernel, the regularization that appears in kernel ridge regression is redundant, and we keep it constant.

## H  GENERALIZED WIGNER ITERATIONS

Even though one cannot compute element-wise non-linear functions of kernels with $\lambda \neq 0$ without disrupting their equivariant behavior, it is possible to define equivariant non-linear functions of the

kernels through their Taylor expansion, e.g.,

$$\exp\left(k_{\mu\mu'}^{\lambda\sigma}\right) \equiv \sum_{n=0}^{\infty} \frac{1}{n!} k_{\mu\mu'}^{n,\lambda\sigma} . \tag{20}$$

Much as it is the case for matrix functions, one can apply several tricks to evaluate these quantities more efficiently than through a truncated series expansion. For example, one can evaluate the exponential through a scaling-and-squaring relation

$$\exp\left(k_{\mu\mu'}^{\lambda\sigma}\right) = [\exp\left(k_{\mu\mu'}^{\lambda\sigma} /2^p\right)]^{2^p}. \tag{21}$$

One first computes $\exp\left(k_{\mu\mu'}^{\lambda\sigma} /2^p\right)$ with a low-order expansion (which works because $2^p$ makes the argument of the exponential very small) and then apply the generalized Wigner iteration $p$ times, multiplying each time the result by itself.

In a similar spirit, if one is only interested in the calculation of all *invariant* kernels up to $\nu = \nu_{\max}$, the Wigner iteration procedure can be simplified. Indeed, it is sufficient to perform full (equivariant) Wigner iterations only up to $\lceil \nu_{\max}/2 \rceil$ and then combine low-order equivariant kernels to get high-order invariant kernels. For example, if $\nu_{\max}$ is even, $k_{00}^{\nu_{\max},01}$ can be calculated as the product of the $k_{\mu\mu'}^{\nu_{\max}/2,\lambda\sigma}$ kernels with themselves via an inexpensive invariant-only Wigner iteration. Due to the $\lambda \leq \lambda_{\max}$ truncation strategy, these kernels might not exactly correspond to those calculated via full Wigner iterations. However, we did not find any differences in performance between the two evaluation strategies, which simply correspond to slightly different angular truncations of the high-order kernels.

## I  HYPERPARAMETERS

### I.1  WIGNER KERNELS

When it comes to the usability of a model, a distinction should be made between "convergence" hyperparameters and "optimizable" hyperparameters. The former are those that show a monotonic improvement of the accuracy of the model as they are increased, but which need to be set to a finite value for practical feasibility. The question then becomes whether they can be converged without compromising the computational speed of the model. In the Wigner kernel case, these are $r_{\mathrm{cut}}$, $\nu_{\max}$, $\lambda_{\max}$, and $n_{\max}$.

- $r_{\mathrm{cut}}$ only enters the initial calculation of the $\nu = 1$ kernels. As a result, the model's training and evaluation times are virtually unaffected by its choice, as long as it is within the typical values for short-range interactions (up to approximately 10 Å).

- The same is true for $n_{\max}$, i.e., the number of radial basis functions used to calculate the $\nu = 1$ kernels: as it does not enter the Wigner iterations, it can be converged almost arbitrarily. Using a Laplacian eigenstate radial basis (Bigi et al., 2022), we did not notice any significant improvement to the accuracy of the models past $n_{\max} = 25$, hence we set it to that value for all benchmarks.

- In contrast, the number of Wigner iterations needed to evaluate the kernels grows linearly with $\nu_{\max}$. We did not find $\nu_{\max}$ to limit the accuracy or the computational cost of the Wigner kernel model in any of our benchmarks.

- $\lambda_{\max}$ is the most critical of these convergence hyperparameters, as the computational performance of the proposed model depends heavily on it. Although going past $\lambda_{\max} = 3$ or $4$ is impractical with our current implementation, our results do not identify this limitation as critical to improve the accuracy of the model. We provide a tentative explanation of this phenomenon in Section 3.2 and Appendix J.

The convergence hyperparameters used in this work are reported in Table 3.

In addition, the Wigner kernel model as presented in this work has two *optimizable* hyperparameters. These are the $C$ and $r_0$ coefficients that enter the density definition in Eq. (16). Given their very

small number, we optimize these via a grid search. The resulting optimized hyperparameters are available in Table 4. The fact that, in the formulation we present here, two physically interpretable density modulation parameters unequivocally determine the value of the kernels is a significant advantage of our framework. As discussed in the main text, the exponential-like kernel parameters can be optimized without having to re-compute the kernels, and we optimize them automatically by cross-validation within the training set. Finally, the ration of the weights of the forces and energies in the (quadratic) loss function is always set to 0.03 for the rMD17 dataset.

| Model | $r_{\text{cut}}$ (Å) | $\nu_{\text{max}}$ | $\lambda_{\text{max}}$ |
|---|---|---|---|
| Methane | 6.0 | 4 | 0, 1, 2, 3 |
| Gold | 6.0 | 2, 3, 4, 6 | 3 |
| rMD17 | 10.0 | 4 | 3 |
| QM9 | 5.0 | 4 | 3 |

Table 3: "Convergence" hyperparameters used for the WK models in Sec. 3.

| Model | $C$ (Å) | $1/r_0$ (1/Å) |
|---|---|---|
| Methane | 0.6 | 0.2 |
| Gold | 0.6 | 0.2 |
| rMD17 - Aspirin | 0.2 | 1/2.8 |
| rMD17 - Azobenzene | 0.35 | 1/5.0 |
| rMD17 - Benzene | 0.4 | 1/5.8 |
| rMD17 - Ethanol | 0.25 | 1/3.4 |
| rMD17 - Malonaldehyde | 0.3 | 1/3.4 |
| rMD17 - Naphthalene | 0.3 | 1/6.3 |
| rMD17 - Paracetamol | 0.25 | 1/3.3 |
| rMD17 - Salicylic acid | 0.3 | 1/4.2 |
| rMD17 - Toluene | 0.3 | 1/4.9 |
| rMD17 - Uracil | 0.4 | 1/4.1 |
| QM9 - all targets | 0.03 | 1.0 |

Table 4: "Optimizable" hyperparameters used for the WK models in Sec. 3.

### I.1.1 SOAP-GPR AND LINEAR SOAP

In our benchmarks, we also provide fits for the SOAP-GPR and linear SOAP methods. The SOAP descriptors (Bartók et al., 2013) present a large number of hyperparameters. In the case of SOAP-GPR, these need to be added to the choice of the kernel (Deringer et al., 2021). With such a large hyperparameter space, it is almost mandatory to rely on previous knowledge and common practice. Hence, for the random methane dataset, we employed a GTO basis with $l_{\text{max}} = 6$, $n_{\text{max}} = 8$, a Gaussian smearing of 0.2 Å which was found to be optimal in Pozdnyakov et al. (2021), and the same radial scaling that was used for the QM9 dataset in Willatt et al. (2018). For the gold cluster dataset, we used the same SOAP hyperparameters that were used in the silicon fit in Goscinski et al. (2021). A squared kernel was used in all SOAP-GPR models, as it is one of the most common choices (Deringer et al., 2021).

### I.1.2 LE-ACE

In this work, LE-ACE was benchmarked on the gold cluster dataset. The LE-ACE model as presented in (Bigi et al., 2022) has $\nu_{\text{max}} + 1$ hyperparameters, roughly corresponding to the maximum Laplacian eigenvalues for $\nu = 1, ..., \nu_{\text{max}}$ plus a radial transform parameter. Since we used $\nu_{\text{max}} = 6$, we found the resulting hyperparameter space to be impossible to optimize rigorously and we therefore optimized it heuristically. We suspect that the successful use of exponential kernels in this work will provide valuable insights in designing more compact and effective hyperparameter spaces for models such as ACE.

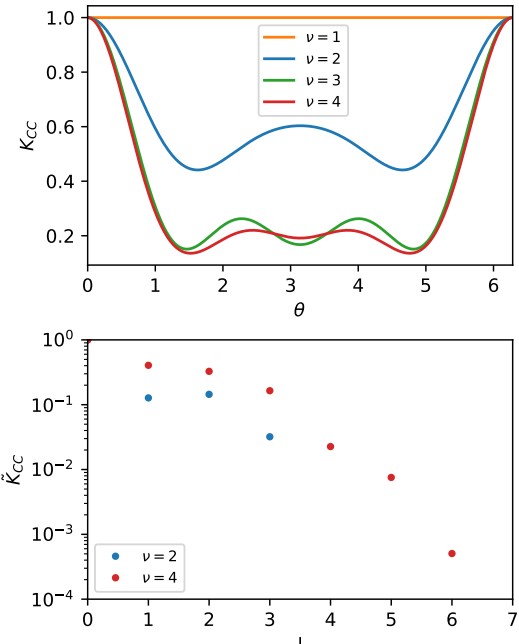

Figure 4: Top panel: angular scan showing a carbon-centered kernel between a $CH_4$ molecule and a $CH_2$ molecule. The $CH_4$ molecule is a random molecule from the methane dataset, while the $CH_2$ molecule has C-H distances of 1.0 and 1.5 Å, and the H-C-H angle $\theta$ is free to rotate. Bottom panel: Fourier transform coefficients of the curves in the top panel at frequencies $l/2\pi$, showing that, although $\lambda_{\max} = 3$ for both kernels, the $\nu = 4$ kernel contains higher ($l > 3$) frequency components.

### I.1.3 MACE

MACE was exclusively used to obtain its timings on rMD17. For consistency with the rMD17 results in Batatia et al. (2022b), $L_{\max} = 3$ and $n_{\text{channels}} = 256$ were used.

## J ANGULAR SCANS AND KERNEL RESOLUTION

To demonstrate the increase in resolution afforded by high-$\nu$ kernels, we compute C-centered Wigner kernels between a random $CH_4$ environments and a set of $CH_2$ structures where we vary the $H{-}C{-}H$ angle ($\theta$) for fixed $C{-}H$ distances. This experiment reveals how higher-$\nu$ kernels are capable of describing higher-frequency components of the $H{-}C{-}H$ angular correlations (Fig. 4). Thus, body-order and structure-space resolution are not fully decoupled.

## K STATISTICAL SIGNIFICANCE

Figures 5 and 6 show the standard deviation and standard errors, respectively, of our learning curves. Since training of our kernel and linear methods is deterministic, such variations exclusively result from different train/test splits within the respective datasets. Hence, they reflect properties of the four datasets more than properties of the Wigner kernel model or the other tested models.

From Fig. 5 (standard deviations), it is clear that individual fits are not guaranteed to accurately assess the relative performance of the examined models, especially on the random methane and gold datasets. However, Fig. 6 (standard errors) shows that all our averaged results, which were run ten times on different splits, are statistically significant.

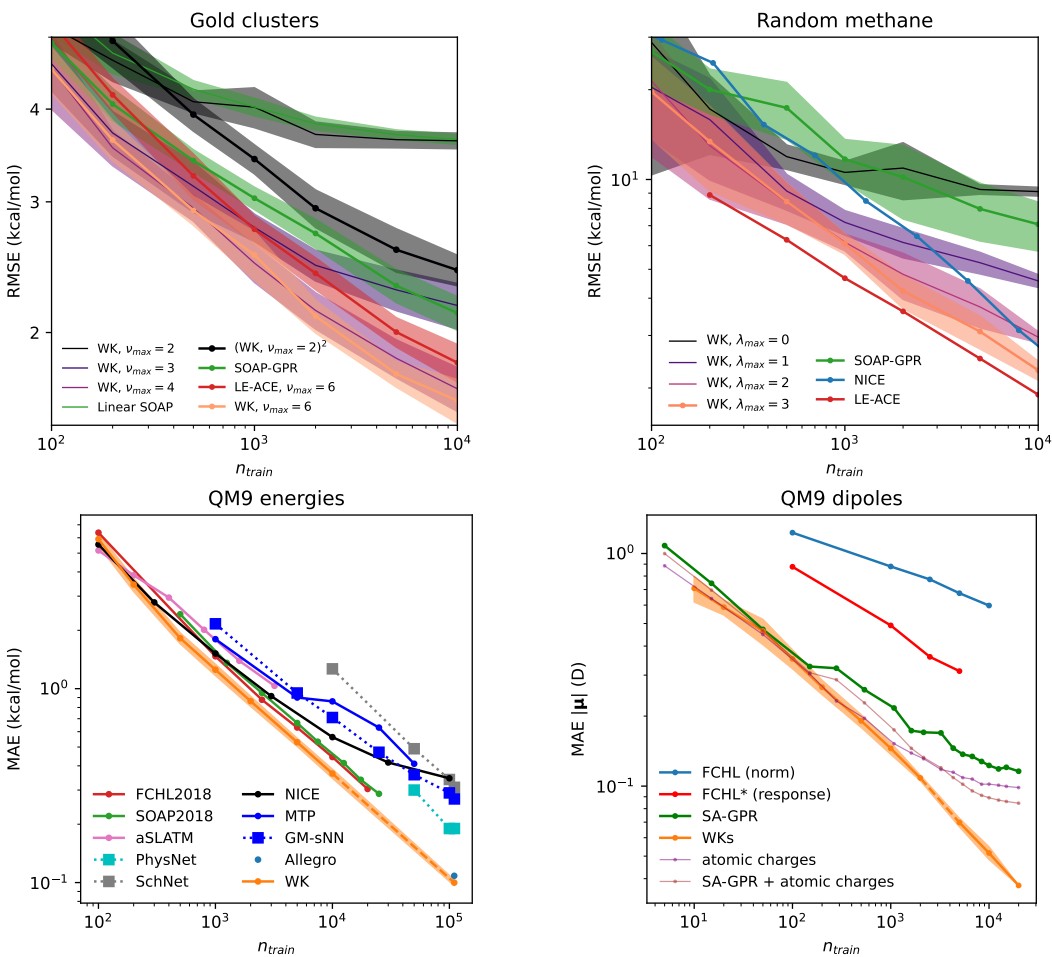

Figure 5: Standard deviations of the learning curves shown in the main text. We do not plot error bars for calculations reproduced from previous work, unless available.

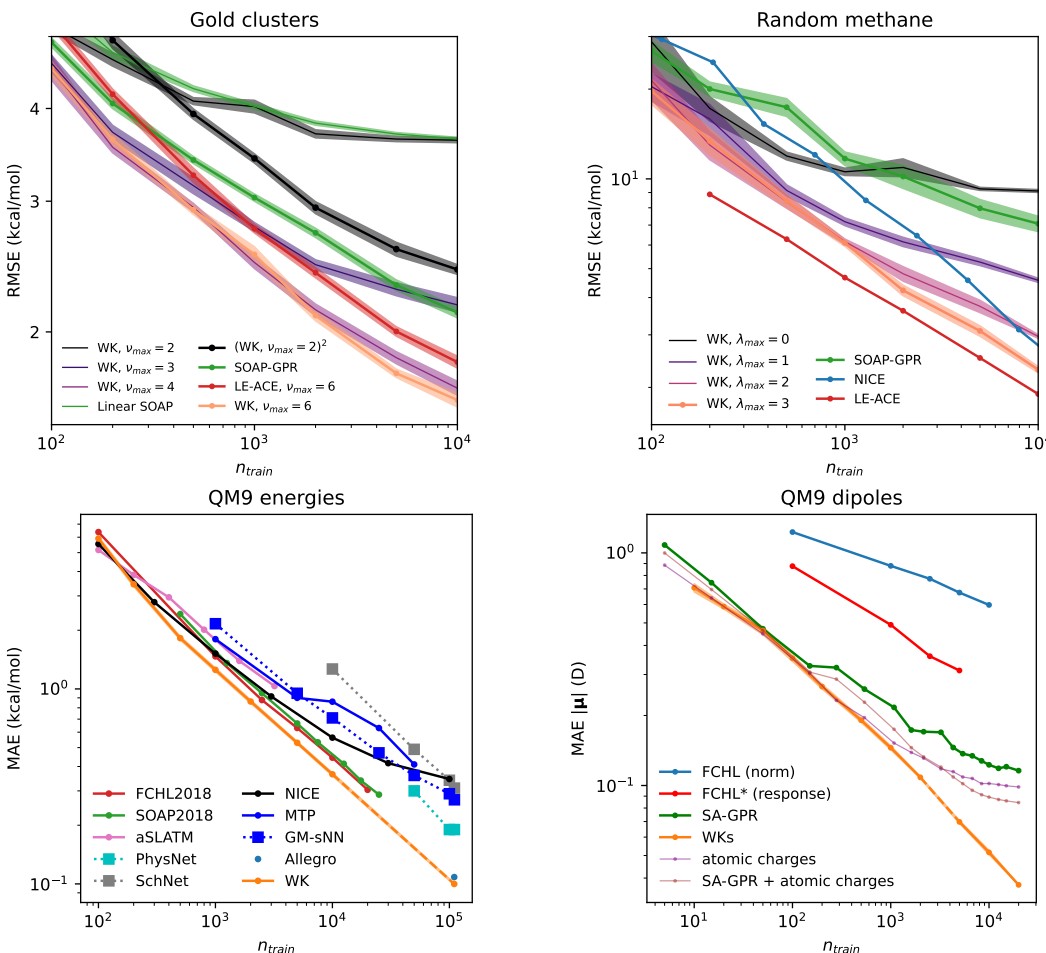

Figure 6: Standard errors of the learning curves shown in the main text. The standard errors for both QM9 figures are hardly noticeable to visual accuracy. We do not plot error bars for calculations reproduced from previous work, unless available.

## L   CODE AVAILABILITY

The code used to generate the results for the Wigner kernel model is available in the Supplementary Material, as well as at https://doi.org/10.5281/zenodo.7952084.

