# OpenReview forum: "Wigner kernels: body-ordered equivariant machine learning without a basis"
_ICLR.cc/2024/Conference — Submitted to ICLR 2024_

### Official Review · Reviewer_VgHZ · 2023-10-27

**Soundness:** 3 good
**Presentation:** 2 fair
**Contribution:** 3 good
**Rating:** 6
**Confidence:** 3

**Summary:**

The authors consider a density-based method for point-cloud representation. They propose a iterative method, called Wigner iteration, to compute a positive definite kernel. The computational cost grows linear with respect to the body-order.

**Strengths:**

The topic is relavant to kernel methods for point-cloud. The proposed method is also related to the representation of interactions between elements, which is important for various applications.

**Weaknesses:**

My main concern is the clarity of the paper.
- Without seeing Eq. (10) in Appendix, I could not understand the role of $\lambda$. Since the authors discuss the computational costs with respect to $\lambda$ in the main text, I think they should clarify what is $\lambda$ in the main text. It is the index of a basis functions in my understanding.
- In addition to $\lambda$, the role of $\mu$ is not clear for me, either. Without knowing what the indexes stand for, we cannot evaluate the computational cost of the sum appeared in Eq. (6). The authors explain that in Appendix E, but it should be in the main text since the computational cost is a crucial topic in this paper.

**Questions:**

For the experimnts in Seciton 4.1, what about the computational time for each case in practice?

---

> ### Author Response · Authors · 2023-11-15
>
> We thank the reviewer for their feedback.
>
> Following the review, we have decided to make significant changes to our manuscript to make it clearer and more reader-friendly. We have defined all terms, provided a table of symbols as an appendix, and included a figure that compares our method to the symmetrized-tensor-product operations that underlie feature-space models as well as equivariant NNs.
>
> It is correct that $\lambda$ is a basis function index, and the same is true for $\mu$. These indices are fundamental to work with spherical tensors, and they stem from the theory of irreducible representations of the SO(3) group. In particular, $\lambda \geq 0$ indexes different irreducible representations, while $-\lambda \leq \mu \leq \lambda$ indexes the different basis functions (or components) within each irreducible representation. We believe that our new table of symbols will greatly help the readers of our work in this regard.
>
> We agree with the reviewer that computational cost considerations are very important in chemistry and materials science applications. We have moved most of the computational cost analysis to the main text, although the detailed derivation of the scaling laws remains in an appendix due to the page constraints.
>
> As a final note, we have made clearer links between our method and its potential application in 3D computer vision [1], where objects are most often represented as point clouds. We have also highlighted relationship between our contribution and the statistical equivalence between infinitely-wide neural networks and Gaussian processes [2]. This showcases the complete nature of our new representations, as opposed to the truncated versions that have been used previously to our work.
>
> Regarding the question by the reviewer, the timings are relatively uniform for both MACE and the Wigner kernels across all molecules, as they are all of similar size. We would like to stress that these timings are only indicative, that they are implementation-dependent and that, in the WK case, they can be improved by more than an order of magnitude by making the active points atomic environments instead of full structures.
>
> [1] Qi, Charles R., et al. "Pointnet: Deep learning on point sets for 3d classification and segmentation." Proceedings of the IEEE conference on computer vision and pattern recognition. 2017.
>
> [2] Lee, Jaehoon, et al. "Deep Neural Networks as Gaussian Processes." International Conference on Learning Representations. 2018.

---

> > ### Comment · Reviewer_VgHZ · 2023-11-23
> >
> > Thank you for your response and the revision. Although I'm still a little concerned about the mathematical notations, the revision may help readers understand the paper more clearly. I will keep my score.

---

> ### Author Response · Authors · 2023-11-23
>
> We thank the Reviewer for their reply and for their feedback, which substantially helped improve the quality of the paper.

---

### Official Review · Reviewer_BHT2 · 2023-11-01

**Soundness:** 2 fair
**Presentation:** 2 fair
**Contribution:** 2 fair
**Rating:** 5
**Confidence:** 2

**Summary:**

The paper focuses on atomic-scale description of molecules and materials using point-cloud representations of physical objects. The authors propose a novel density-based approach using "Wigner kernels."

**Strengths:**

1. The proposed Wigner kernel affords computationally efficient density-based representation.
2. Comprehensive empirical results are provided demonstrating the utility of the Wigner kernel on various datasets.

**Weaknesses:**

1. I found the paper hard to follow with many terminologies in 3D objects or molecules context. Perhaps the authors can improve the presentation by highlighting the ML related contributions.
2. The formulation Eq (4) seems to be limited to only three-dimensional problems.
3. The formulation Eq (1) is highly related to the kernel mean embedding (KME) (see, e.g., Muandet et al., Kernel Mean Embedding of Distributions: A Review and Beyond). However, the related work does not compare with any of the KME literature.

**Questions:**

See above.

---

> ### Author Response · Authors · 2023-11-15
>
> We thank the reviewer for their comments and questions.
>
> Here are our replies to the three questions, in the same order:
>
> We thank the reviewer for pointing this out. We have revised our manuscript to highlight the relevance, connections and applications of our contribution within the machine-learning literature. These include the link with the statistical equivalence between neural network and Gaussian processes, as well as many connections to three-dimensional computer vision. As for the presentation itself, we have made significant efforts to make the paper more reader-friendly. In particular, we provided a definition for all the employed notation, added an appendix with a comprehensive table of symbols, and illustrated the contrast between the Wigner kernels and feature-space models in a new figure.
>
> The reviewer is correct. However, it would be relatively simple to generalize our formulation to arbitrary dimensions and symmetry groups. For example, it would be possible to apply our method to 2D point-cloud classification simply by replacing the SO(3) Wigner-D matrices by their SO(2) counterparts and applying a Gaussian process classifier model. This would be trivial to implement and very computationally efficient, as all irreducible representations of SO(2) are one-dimensional. Our paper is voluntarily focused on the 3D case, as this is relevant to atomic configurations. In addition to this, while 2D machine learning inputs are usually represented as images composed of pixels, 3D objects are most often represented as point clouds [1]. This makes our model applicable to classification on point clouds, either as a Gaussian process classifier, or through the inclusion of the proposed Wigner iteration as a layer in an equivariant neural network architecture. As a side note, we think it should be possible to modify our construction to be applied to 2D images composed of pixels through some sort of coarse-graining. This would provide formally complete and efficient geometric representations for 2D image recognition.
>
> Using Eq. 1 is common practice when describing molecules with kernel methods, and it is usually motivated on physical grounds. What the reviewer suggested is an interesting connection that we were not aware of. It is correct that the similarity measure between two structures, as formulated in Eq. 1, is equivalent to the kernel mean (sum, in this case) embedding of the kernels defined between individual atomic neighborhoods. We thank the reviewer for pointing this out, and we have added this connection to the paper, along with its reference.
>
> [1] Qi, Charles R., et al. "Pointnet: Deep learning on point sets for 3d classification and segmentation." Proceedings of the IEEE conference on computer vision and pattern recognition. 2017.

---

### Official Review · Reviewer_LkVM · 2023-11-01

**Soundness:** 3 good
**Presentation:** 2 fair
**Contribution:** 2 fair
**Rating:** 6
**Confidence:** 2

**Summary:**

The paper describes a kernel that is equivariant w.r.t. SO(3) and accounts for neighbor information "body order". The kernel is computed using an iterative computation of lower body-order kernel values. Using kernel ridge regression, the paper trained the Wigner kernel model on gold clusters, random methane configurations energies, QM9 molecule energies, dipoles, and rmd17 dataset energies and forces, and compares with several other models.

**Strengths:**

quality+clarity: Good set of experiments. Authors tried 4 datasets and compared with many additional models. Figures and tables are high quality.
originality: I'm not familiar with literature to tell if kernel in eq 5 is novel. The iterative calculation of the kernel and application to molecule properties is novel.
significance: Kernel ridge regression with the developed kernel outperforms SOTA models in several cases.

**Weaknesses:**

It is not entirely obvious to me how this could be applied outside of molecule predictions. Perhaps authors can add a bit of description here?

**Questions:**

-can authors define terms in equation 2?

"this formulation of the high-order kernels is entirely lossless in terms of the radial basis and the dimension of composition (chemical element) space."(pg 4)
-do the kernels take into account the atom composition?

-can authors include a comparison of training time across the models, for ex. the x-axis of Figs 1 and 2?

---

> ### Author Response · Authors · 2023-11-15
>
> We thank the reviewer for their positive comments.
>
> We have revised our manuscript to make the presentation more accessible to a machine-learning audience, and to make links between our result and other machine-learning results and/or applications clearer.
>
> To expand on the latter point, we believe there are three main directions in which our complete representations for 3D point clouds can be employed outside of atomistic and molecular systems:
> - **Classification tasks**: although we showed results for regression, which are the only relevant machine learning exercise in the prediction of microscopic properties, the Wigner kernel formalism could have a great impact in 3D object recognition [1], as 3D objects are most often represented as point clouds. This would be achievable through the use of a Gaussian process classifier [2] with a covariance function corresponding to the Wigner kernels.
> - **Incorporation inside neural network architectures**: the Wigner iteration can be used as a layer in an equivariant neural network architecture, which would offer a more practical alternative to Gaussian process regression/classification. We believe this to be a very promising route to achieve richer and more complete representations in geometric machine learning.
> - **Generalization to other groups and dimensions**: although our formalism is presented for the SO(3)/O(3)/E(3) groups, it is easily generalizable to other groups. In particular, the 2D implementation of the Wigner iteration is trivial, as the irreducible representations of the SO(2) group are all one-dimensional.
>
> As a result, we believe that the proposed method could have a large impact on all areas of geometric machine learning.
>
> Moving on to the questions:
> - All the terms have now been defined in Eq. 2. We have also added a table of symbols in the appendices, which will be very helpful for the readers of our work who are not familiar with the SO(3) group formalism.
> - Yes, they do. In the initial version, we decided to omit the chemical species indices in some key equations to make the presentation tidier. However, we realize that this might be confusing and we have now added them back. In particular, Eq. 5 shows that kernels between two individual atoms are symmetrized sums of integrals between densities of the same kind (that is, belonging to the same neighboring atom species). This amounts to considering each chemical element as its own dimension in the chemical species space, and it is the kernel equivalent of one-hot encoding of the chemical species of the neighbor atoms.
> - We do not feel comfortable in providing training (or inference) timings for the models in Figs. 1 and 2 directly in our manuscript, even though they are typically larger than those for the Wigner kernel. This is because we implemented those models ourselves to ensure full consistency with the Wigner kernels, while more optimized (and often less accurate) packages exist. In many cases such implementations make approximations to enhance computational efficiency and usability, and they are therefore unsuitable for careful ablation and comparison tests. However, we can reassure the reviewer that, in the data regime we typically work on, training times are considered to be a significant strength of kernel models as, once the kernels are available, the fit consists of a single linear algebra operation. In our ablation studies, the training times of our method were comparable to those of other kernel and linear models, and much faster than those of neural networks trained on the same datasets. Of course, as we mention in the manuscript, increasing the train-set size requires the use of a projected-process approximation of the kernel, which we do not explore here as it would obfuscate the analysis of the converged-basis limit of body-ordered equivariant models by introducing the active space size as a further convergence parameter.
>
> As a final note, the kernels in Eq. 5 are not completely novel, as they had been considered in Ref. [3]. Despite the fact that they had been known for a decade, nobody had computed these high-order kernels explicitly previously to our work due to their very high computational cost. The focus shifted instead into providing good approximations to the real-space density correlations that enter the kernels. Our Wigner iteration allows to compute these complete representations of geometric structures, showing that they indeed provide a very rich and complete representation of 3D objects.
>
> [1] Qi, Charles R., et al. "Pointnet: Deep learning on point sets for 3d classification and segmentation." Proceedings of the IEEE conference on computer vision and pattern recognition. 2017.
>
> [2] Williams, Christopher KI, and Carl Edward Rasmussen. Gaussian processes for machine learning. Vol. 2. No. 3. Cambridge, MA: MIT press, 2006.
>
> [3] Bartók, Albert P., Risi Kondor, and Gábor Csányi. "On representing chemical environments." Physical Review B 87.18 (2013): 184115.

---

### Official Review · Reviewer_hS7z · 2023-11-01

**Soundness:** 3 good
**Presentation:** 2 fair
**Contribution:** 2 fair
**Rating:** 5
**Confidence:** 4

**Summary:**

Decorated 3D point-cloud representations provide a way to describe molecules and take into account the complexity of the interactions between atoms. Kernels are especially well suited to include by design wishable properties such as equivariance with respect to symmetry operations in the group of 3D rotations SO(3). In particular, symmetry-adapted kernels under the form of body-ordered kernels have been proposed in the literature. However computing these kernels with high order \nu > 2 is impractical. This paper proposes a single to compute iteratively high-\nu kernels by relying on lower-order kernels. This computation is backed up by a proof given in Appendix. A numerical section is devoted to test the so-called Wigner kernels in the context of  Kernel Ridge Regression on datasets where a high body-order is needed (random methane dataset, QM9 and RM17 datasets).

**Strengths:**

Soundness: In the context of molecule representation, the paper focuses on body-ordered kernels and proposes a new way to compute it based on them. The contribution undoubtedly opens the door to application of these kernels in atomistic properties prediction tasks. Moreover even if it does not insist on that point, it also gives a nice example of geometrical learning where the hypothesis space induced by a kernel inherits by definition of geometrical properties.

Originality: While this work follows in the footsteps of many recent works about kernels for molecules based on 3D point-cloud presentations and symmetry-adapted kernels, it proposes a new way of computing them and as so, is original.
Clarity: the paper awfully lacks of clarity for a machine learning reader (see weaknesses). The general message which is not too complex in itself is made noisy by a lot of implicit statements.

**Weaknesses:**

Soundness: Overall the contribution is very technical in terms of chemistry and less informative in terms of machine learning. I assume that it will be of limited interest for a vast majority of the community in ML and would be more highlighted in a dedicated venue.

Presentation: This paper suffers a lot from its presentation. It seems to me that the writing was intended for chemists and not the machine learning community.  It cruelly lacks of definitions and notations. The reader has to read a few chemistry papers to get a clear idea of each notation: for instance in equation (3),  which space does A_i belong to ?, recall what is a Wigner D-matrix, in equation (4), define x and r. More generally please start by defining how you describe a molecule ( a set of 3D coordinates and a set of associated labels, distances between atoms, forces ??) - The numerical experiment section suffers from the same default: the comments reflect a high level of expertise in chemistry from the authors but fail to highlight the interest for machine learning.

The lack of information about definitions and notations also prevents from a careful analysis of the computations at work here: what is the (analytical) complexity in time ? memory requirements...

**Questions:**

(1) please define and provide your notations at eahc step of your paper (after related works)
(2) please clearly explain how body-ordered kernels were computed so far and compare the analytical complexity in time with those of the Wigner kernel.
I've read the rebuttal and raised consequently my score.

---

> ### Author Response · Authors · 2023-11-15
> **General comments**
>
> We thank the reviewer for their comments and suggestions.
>
> It is true that we highlighted many contributions to the field which are related to our construction. However, despite these relationships, our approach does not simply afford a speed-up of an established method. Indeed, although they had been known since 2013 [1], what we call “Wigner kernels” had never been previously computed or tested past 𝜈 = 2 due to their computational inconvenience. Even in the 𝜈 = 2 case, significant approximations are usually made [2, 3].
>
> For context, 𝜈 = 4 or higher is typically needed to "converge" to a satisfactory accuracy (in our paper, this can be seen from the ablation study on gold clusters). Given these limitations of the current models, researchers in the field have opted for one of two methods:
> - using arbitrary functions of (approximate) 𝜈 = 2 kernels that do not span the full space of geometric correlations [4];
> - computing heavily truncated feature-space representations, either explicitly or via the use of equivariant neural networks.
>
> Our results section effectively compares the Wigner kernels to these approaches.
>
> We realize that our initial presentation and result discussion were chemistry-focused and perhaps less adapted a machine learning audience. We thank the reviewer for pointing this out, and we have made significant changes to our manuscript. Most notably, we have defined all terms as suggested, provided a comprehensive table of symbols as the first appendix, and helped the reader visualize the contrast between our method and explicit or implicit feature-space approaches with a new figure. We have also limited the amount of chemical details in the discussion of the benchmarks, highlighting instead our more fundamental contributions.
>
> Although we fully agree with the reviewer that our initial presentation was lacking, we believe that our contribution could be very valuable to the theory of learning geometric representations. We establish an equivalence between kernel methods and infinitely-wide linear/NN models, which can be seen as a geometric equivalent to the well-known statistical relationship between neural networks and Gaussian processes [5].
>
> Drawing from this theoretical result, we provide an implementation of such complete representations and empirical results on some of the most popular benchmarks in the field. In this way, we allow future research to compare with and/or compute complete body-ordered empirical results, which are independent of the choice of truncated representations (and often more accurate).
>
> As we discuss in the paper, as well as in our reply to reviewers LkVM and BHT2, our kernels could also be used for classification of point clouds, simply by applying a Gaussian process classifier method [6]. Shifting the formalism from the E(3)/O(3) group to other symmetry groups would also be relatively simple, and it would further allow our complete representations to be applied to problems with different symmetries and/or dimensionalities. The inclusion of the Wigner iteration as a layer of an equivariant neural network architecture is also a very promising direction.
>
> Finally, the computational efficiency of our method allows it to be very impactful in practical applications where latency is a primary concern, and especially in chemistry and materials science.
>
> [1]  Bartók, Albert P., Risi Kondor, and Gábor Csányi. "On representing chemical environments." Physical Review B 87.18 (2013): 184115.
>
> [2] Musil, Felix, et al. "Physics-inspired structural representations for molecules and materials." Chemical Reviews 121.16 (2021): 9759-9815.
>
> [3] Deringer, Volker L., et al. "Gaussian process regression for materials and molecules." Chemical Reviews 121.16 (2021): 10073-10141.
>
> [4] Pozdnyakov, Sergey N., et al. "Incompleteness of atomic structure representations." Physical Review Letters 125.16 (2020): 166001.
>
> [5] Lee, Jaehoon, et al. "Deep Neural Networks as Gaussian Processes." International Conference on Learning Representations. 2018.
>
> [6] Williams, Christopher KI, and Carl Edward Rasmussen. Gaussian processes for machine learning. Vol. 2. No. 3. Cambridge, MA: MIT press, 2006.

---

> ### Author Response · Authors · 2023-11-21
> **Answers to the questions**
>
> [Edit: since we have expanded on our general comments, we are moving our answers to the questions to a different reply]
>
> Here are our answers:
>
> 1. We thank the reviewer for pointing this out. We have made significant revisions to the manuscript, providing explicit definition for all the quantities used in the main text. We have also added a list of symbols in the appendix and included a brief discussion of the feature-space iteration that is at the basis of equivariant neural networks. Finally, we have highlighted further implications of our work for machine learning on 3D point clouds.
>
> 2. Low-order kernels ($\nu=1,2$) are generally computed as described in Section 3 by performing scalar products of density-correlation (ACE/NICE) features. It is not current practice to compute higher-order kernels because of their prohibitive cost. In other words, previously to our work, kernel methods had the exact same exponential scaling as the ACE/NICE methods discussed in the scaling section. That is, $\mathcal{O}((a_\mathrm{max}n_\mathrm{max})^{\nu})$, where $a_\mathrm{max}$ is the number of chemical species in the dataset and $n_\mathrm{max}$ is the number of $\nu=1$ radial basis functions used in the discretization. As a result of this unfavorable scaling, current models operate in feature space, performing heavy approximations to keep complexity under control. Our contribution is to show that operating in kernel space affords an alternative way to increase the body order that avoids the exponential scaling. That is, Wigner kernels scale as $\mathcal{O}(a_\mathrm{max}n_\mathrm{max})$, only due to the calculation of the $\nu=1$ kernels. The scaling with respect to $\nu_\mathrm{max}$ is linear ($\mathcal{O}(\nu_\mathrm{max})$), since the calculation of kernels of order $\nu_\mathrm{max}$ requires $\nu_\mathrm{max}-1$ Wigner iterations. In addition, the resulting representations are complete, and they usually perform better than truncated methods as a result. The drawback is an increase of the polynomial complexity in the angular parameter $\lambda_\textrm{max}$: while ACE/NICE scale as $\mathcal{O}(\lambda^5_\textrm{max})$, Wigner kernels scale as $\mathcal{O}(\lambda^7_\textrm{max})$. However, recent results in the field demonstrate that only a small $\lambda_\textrm{max}$ (2 or 3) is necessary to converge in accuracy. We investigate this phenomenon in the methane ablation study and in an appendix, establishing that this is due to self-correlations in the atomic positions that effectively incorporate high-$\lambda$ behavior in low-$\lambda$ representations. We have made the presentation of our scaling laws clearer in the main text. For more detailed information, the reviewer can refer to Appendix F.

---

> ### Comment · Reviewer_hS7z · 2023-11-23
> **Feedback on the rebuttal**
>
> I think the authors have improved the paper in terms of clarity and I thank them for the major revision.

---

> > ### Author Response · Authors · 2023-11-23
> >
> > We thank the Reviewer for their comments and feedback, which have undoubtedly improved the manuscript in many ways.

---

### Author Response · Authors · 2023-11-15
**Reply to all Reviewers**

We thank the reviewers for their replies and questions. We are very pleased with their general comments, which highlight the soundness and quality of our model, as well as our empirical results.

We see that a common concern, shared by three reviewers, pertains to the presentation of our work, which relied too heavily on previous results that are relatively well-known within the chemical ML community, but probably not as much in the community contributing to ICLR. We believe that the best way to address these comments is to heavily revise the presentation of our manuscript: we discuss how we did so in the individual responses, and we highlight the changes in the manuscript (mark-up will be removed at the end of the discussion session).

Our general approach is to make the manuscript more self-contained, briefly explaining the state of the art of equivariant models in chemistry before commenting on the significance of our contribution - which is to achieve the complete-basis limit of such models while avoiding an exponential scaling of the cost with the body order of the kernels. We also discuss more clearly the relationship of our results with generic geometric learning and with equivariant deep-learning models. These are based on the same Clebsch-Gordan product operation as descriptor-based methods, and they also require a careful truncation of the basis in their current implementations. In this respect, we also want to stress that the significance of our contribution goes beyond the computational efficiency and the benchmark performance that we demonstrate. Wigner kernels can be seen as a way to extend the established equivalence between infinitely wide neural networks and Gaussian processes to a geometric learning context. In this way, Wigner kernels allow probing the infinite-width limit of several shallow and deep geometric learning frameworks, making our contribution, in our opinion, very well fitting for a conference dedicated to the learning of representations.

Finally, we have added a dedicated table of symbols in the appendix to help in the reading of our work.

We are very interested in receiving additional feedback, especially on whether we have satisfactorily addressed the reviewers’ concerns on presentation, and we are more than willing to elaborate further on any point that might still be unclear.

---

### Author Response · Authors · 2023-11-22
**Removing Mark-up**

Since the end of the discussion window is approaching, we are removing the mark-up. The marked-up version is still available in the history of the submission. Compared to the final version, it only lacks some minor changes to the "scaling and computational cost" sub-section that we have made to better address the reviewers' recommendations. We hope our replies have succesfully clarified any ambiguities and addressed the reviewers' concerns.

---

### Meta-Review · Area_Chair_BH1W · 2023-12-07

**Metareview:**

This paper proposes a density-based method for point cloud representation, which involves computing Wigner kernels that can be carried out iteratively. Several chemical applications are presented.

While reviewers generally appreciate the approach for computing the Wigner kernel and the chemical applications, they also generally have trouble reading the paper, due to the many notations from chemistry/physics. The theoretical derivation given in the Supplementary Material likewise relies heavily on papers in chemistry and physics.

In particular, I agree with Reviewer hS7z that the paper would be of limited interest to the majority of the machine learning community and would be better suited for a venue in chemistry or physics.

**Justification For Why Not Higher Score:**

The paper is not appropriate for the general machine learning audience. It should be submitted to a different venue, e.g. in chemistry or physics.

**Justification For Why Not Lower Score:**

N/A

---

### Decision · Program_Chairs · 2024-01-16

Reject